# Glycolysis on F-18 FDG PET/CT Is Superior to Amino Acid Metabolism on C-11 Methionine PET/CT in Identifying Advanced Renal Cell Carcinoma at Staging

**DOI:** 10.3390/cancers13102381

**Published:** 2021-05-14

**Authors:** Suk-Hyun Lee, Jee-Soo Park, Hyunjeong Kim, Dongwoo Kim, Seung-Hwan Lee, Won-Sik Ham, Woong-Kyu Han, Young-Deuk Choi, Mijin Yun

**Affiliations:** 1Department of Nuclear Medicine, Severance Hospital, Yonsei University College of Medicine, Seoul 03772, Korea; shlee0021@hallym.or.kr (S.-H.L.); KDWOO@yuhs.ac (D.K.); 2Department of Radiology, Hallym University Kangnam Sacred Heart Hospital, Seoul 07441, Korea; 3Department of Urology, Urologic Science Institute, Severance Hospital, Yonsei University College of Medicine, Seoul 03772, Korea; sampark@yuhs.ac (J.-S.P.); LEESEH@yuhs.ac (S.-H.L.); UROHAM@yuhs.ac (W.-S.H.); HANWK@yuhs.ac (W.-K.H.); 4Department of Nuclear Medicine, Yongin Severance Hospital, Yonsei University College of Medicine, Yongin-si 17046, Gyeonggi-do, Korea; KHJ7564@yuhs.ac

**Keywords:** F-18 fluorodeoxyglucose, C-11 methionine, positron emission tomography, renal cell carcinoma

## Abstract

**Simple Summary:**

Alteration of metabolism, including glycolysis and glutaminolysis in malignant tumours, has become a hallmark of cancer and related biological aggressiveness. The metabolic signature of each cancer has been actively investigated for potential new drug development. Of the metabolic imaging biomarkers, F-18 fluorodeoxyglucose (FDG) and C-11 methionine positron emission tomography/computed tomography (PET/CT) are widely studied to evaluate the degree of glucose metabolism and amino acid metabolism, respectively. In this prospective study, we found that both F-18 FDG and C-11 methionine uptakes on PET/CT were heterogeneous in renal cell carcinomas, and increased uptake was associated with higher grades of both radiotracers. Additionally, metabolic tumour volume on F-18 FDG PET/CT but not C-11 methionine PET/CT was significant in predicting advanced-stage renal cell carcinoma. These metabolic features derived with PET/CT may help in the development of new drugs targeting glucose and amino acid metabolic pathways.

**Abstract:**

We evaluated the value of F-18 fluorodeoxyglucose (FDG) and C-11 methionine positron emission tomography/computed tomography (PET/CT) to predict high-Fuhrman grade and advanced-stage tumours in patients with renal cell carcinoma (RCC). Forty patients with RCC underwent F-18 FDG and C-11 methionine PET/CT between September 2016 and September 2018. They were classified into limited (stages I and II, *n* = 15) or advanced stages (stages III and IV, *n* = 25) according to pathological staging. Logistic regressions were used to predict the advanced stage using various parameters, including maximum standardised uptake value (SUV_max_) and metabolic tumour volume (MTV). Receiver operating characteristic analyses were performed to predict high-grade tumours (Fuhrman 3 and 4). On univariate analysis, tumour size, SUV_max_ and MTV of F-18 FDG and C-11 methionine, and Fuhrman grades were significant predictors for the advanced stage. On multivariate analysis, F-18 FDG MTV > 21.3 cm^3^ was the most significant predictor (*p* < 0.001). The area under the curve for predicting high-grade tumours was 0.830 for F-18 FDG (*p* < 0.001) and 0.726 for C-11 methionine PET/CT (*p* = 0.014). In conclusion, glycolysis on F-18 FDG PET/CT and amino acid metabolism on C-11 methionine PET/CT were variable but increased in high-grade RCCs. Increased MTV on F-18 FDG PET/CT is a powerful predictor of advanced-stage tumours.

## 1. Introduction

Renal cell carcinoma (RCC) is the sixth-most common cancer in males and the eighth-most common cancer in women. It is the most lethal urological cancer, with a mortality rate of 25% [1]. Accurate staging of RCC is of utmost importance because the selection of different therapeutic strategies, such as surgery or systemic treatment, depends on the detection of metastatic lesions at a single site or several sites [2]. Preoperative prediction of biologically aggressive RCCs is limited owing to the lack of reliable clinical or radiological criteria. Although multigene assays have been increasingly used for improved prognostication, non-invasive imaging biomarkers can help identify patients with a higher risk of advanced disease for appropriate treatment-related decision-making at staging [3].

Alteration of metabolism is a hallmark of cancer and has various biological implications. For example, aerobic glycolysis is promoted by the activation of many growth factor signalling pathways, such as EGFR, glutaminolysis to protect cancer cells from oxidative stress, and tryptophan metabolism to generate immunosuppressive compounds [4]. Investigation of metabolic signatures in each cancer could be useful in developing new drugs targeting specific metabolic pathways. Of the metabolic imaging biomarkers, F-18 fluorodeoxyglucose (FDG) and C-11 methionine (MET) positron emission tomography/computed tomography (PET/CT) are widely used to evaluate the degree of glucose metabolism and amino acid metabolism, respectively [5]. To date, a significant portion of RCCs is known for low F-18 FDG uptake due to low glycolysis [6]. A meta-analysis on the detection of RCC using F-18 FDG PET reported a pooled sensitivity of 62% (95% confidence interval (CI): 49–74%) [7]. Importantly, RCCs with a high F-18 FDG uptake have been associated with tumour aggressiveness, such as Fuhrman grade [2,8,9,10], phosphorylated Akt and S6 kinase [11], and metastatic lesions [2,12,13]. The roles of metabolic tumour volume (MTV) in glycolysis on F-18 FDG PET/CT in primary RCC have not been evaluated.

Since cellular proliferation is associated with increased protein synthesis, radiolabeled amino acids, including C-11 MET, are used to evaluate various malignant tumours [12,13,14,15,16,17,18,19]. Other than protein synthesis, transamination, ATP generation, nucleotide synthesis, and methylation reactions are closely related to amino acids, including MET [20]. C-11 MET is the most-used radiotracer for detecting and grading cerebral gliomas with high sensitivity, even in non-glycolytic, low-grade tumours [21,22]. Unlike F-18 FDG, it has minimal excretion through the kidneys, which may increase its ability to detect RCCs. Although a small pilot study suggested the potential value of C-11 MET for detecting primary tumours and metastases in patients with RCC [23], further research is limited.

This prospective study aimed to assess the effects of glycolysis on F-18 FDG and amino acid metabolism on C-11 MET PET/CT to predict high Fuhrman grade and advanced-stage tumours in patients with RCC.

## 2. Results

### 2.1. Patient Characteristics

Altogether, 40 patients (21 males and 19 females) were included in this study. The mean age was 59.6 ± 12.5 years, and the mean tumour size was 8.2 ± 2.4 cm. Among the 40 patients, 5 (12.5%) had stage I tumours, 10 (25.0%) had stage II tumours, 16 (40.0%) had stage III tumours, and 9 (22.5%) had stage IV tumours. Fuhrman nuclear grades were low (grades 1 and 2) in 12 patients and high (grades 3 and 4) in 28 patients. On visual analysis, F-18 FDG uptake was positive in 30 tumours, and C-11 MET uptake was positive in 24 tumours (75.0% vs. 60.0%, *p* = 0.109). Inter-observer agreement between two interpreters was good in both F-18 FDG (kappa: 0.678, 95% CI 0.453–0.903) and C-11 MET (kappa: 0.652, 95% CI 0.420–0.884) PET/CT. Patient characteristics according to the American Joint Committee on Cancer (AJCC) stages are summarised in Table 1. Characteristics of individual patients can be found in Appendix A.

### 2.2. Logistic Regression Analysis for the Prediction of Advanced-Stage Disease

The limited stage disease was observed in 15 patients, while 25 patients had advanced-stage disease. Simple logistic regression results showed tumour size, F-18 FDG SUV_max_, F-18 FDG MTV, C-11 MET SUV_max_, C-11 MET MTV, and Fuhrman nuclear grades as significant predictors for advanced-stage disease. Among these factors, F-18 FDG MTV was the only significant predictor of advanced-stage disease in the stepwise logistic regression analysis (Table 2). A primary tumour with an F-18 FDG MTV cut-off > 21.3 cm^3^ was a significant predictor of advanced-stage disease (odds ratio, 102.67; 95% CI, 9.69–1087.62; *p* < 0.001). Representative cases are shown in Figure 1 and Figure 2, respectively.

### 2.3. Comparison of F-18 FDG and C-11 MET Uptake in the Primary Tumour and Histopathological Findings

There was a moderate correlation between the SUV_max_ of F-18 FDG and C-11 MET PET/CT (r = 0.63, *p* < 0.001, Figure 3) for primary tumours. Tumours with high Fuhrman grades showed higher SUV_max_ than low-grade tumours on both F-18 FDG (4.1 ± 2.1 vs. 9.2 ± 6.0, *p* = 0.001) and C-11 MET PET/CT (4.4 ± 2.5 vs. 6.7 ± 2.5, *p* = 0.024) (Figure 4). The mean F-18 FDG SUV_max_ of high-grade tumours was 124% higher than that of low-grade tumours, while the mean C-11 MET SUV_max_ of high-grade tumours was only 52% higher than that of low-grade tumours. In the receiver operating characteristic (ROC) analysis for grading, the area under the curve (AUC) was 0.830 for F-18 FDG SUV_max_ (cut-off > 6.0, *p* < 0.001), while it was 0.726 for C-11 MET SUV_max_ (cut-off > 4.3, *p* = 0.014). The AUC of F-18 FDG SUV_max_ was slightly greater than that of C-11 MET SUV_max_, but the difference was not statistically significant (*p* = 0.262). In contrast to the SUV_max_ of F-18 FDG and C-11 MET, tumour size was not correlated with Fuhrman grade (*p* = 0.315). In addition, the SUV_max_ of both F-18 FDG and C-11 MET was not significantly correlated with tumour size (*p* = 0.688 and *p* = 0.794, respectively).

### 2.4. Comparison of Detection of Extra-Renal Metastases between Conventional Imaging and PET/CT

Three patients had regional lymph node metastases on pathological examination, which could be detected on abdominal CT, F-18 FDG PET/CT, and C-11 MET PET/CT. Nine patients had distant metastases (six with lung metastases, two with bone metastases, and one with lung and bone metastases). All lung metastases were detected on chest CT, F-18 FDG PET/CT, and C-11 MET PET/CT. Bone metastases in three patients were negative or equivocal on conventional imaging but showed positive uptake on both F-18 FDG and C-11 MET PET/CT.

## 3. Discussion

Both F-18 FDG uptake, an imaging surrogate for glycolysis, and C-11 MET uptake for amino acid metabolism on PET/CT generally reflect rapid proliferation and poor differentiation in cancer [24,25,26]. A good correlation between F-18 FDG and C-11 MET uptake has been reported in cerebral gliomas [27]. In this study, we also found a moderate correlation between F-18 FDG and C-11 MET uptake in primary RCC. Both metabolic radiotracer uptakes were higher in high-grade tumours than in low-grade tumours, and the difference in uptake between the high-grade and low-grade tumours was more prominent with F-18 FDG. Regardless, ROC analysis showed that both F-18 FDG SUV_max_ and C-11 MET SUV_max_ were useful for predicting high-grade tumours (AUC: 0.830 vs. 0.726, *p* = 0.262). In contrast, tumour size, a component of T staging, showed no significant correlation with Fuhrman grade. Thus, metabolic parameters, such as F-18 FDG uptake or C-11 MET, seemed more important than tumour size in reflecting the biological behaviour of RCCs.

In aggressive tumours, an altered pattern of glucose metabolism is observed, generally called “the Warburg effect” [28]. While upregulation of genes related to glucose metabolism induces aerobic glycolysis in tumours with sufficient oxygen levels for proliferation, it is an inefficient way to generate adenosine triphosphate than oxidative phosphorylation [29,30]. This altered metabolism requires more glucose; therefore, cancer cells usually upregulate glucose transporters [31]. In RCCs, aggressive tumours are known to show a metabolic shift in which there is downregulation of the TCA cycle genes, decreased levels of AMPK and PTEN protein, upregulation of the pentose phosphate pathway, and glutamine transporter genes, and increased enzyme levels for lipid synthesis [32]. In our study, RCCs showed a wide range of F-18 FDG uptake in the primary tumours, and high F-18 FDG uptake was associated with high Fuhrman grades. It seems that the known metabolic shift noted in aggressive RCCs shares metabolic changes in tumours of the Warburg phenotype. Further studies are needed to determine whether increased glycolysis on F-18 FDG PET/CT is reflective of the known metabolic signatures in aggressive RCC.

In addition to glycolysis, glutaminolysis plays an essential role in the growth of cancer cells [33]. In RCC, glutamine is an important source of fatty acids by reductive carboxylation [34,35] and nucleotide biosynthesis [36]. Despite its importance, no radiolabeled glutamine is available in clinical practice. Instead, radiolabeled essential amino acids are currently used in patients, and C-11 MET is one of the most commonly used amino acid-derived radiotracers. Methylthioadenosine phosphorylase (MTAP), a key enzyme in the methionine salvage pathway, is lost in almost all cancers. Accordingly, cancer cells show increased MET levels for survival and growth [37]. In RCCs, alterations in leucine and tryptophan transport by large amino acid transporters have been reported [4,38]. Since MET shares the same transporter system with leucine or tryptophan [39], C-11 MET PET/CT is a potential candidate for evaluating amino acid metabolism in RCCs. In this study, we found variable C-11 MET uptake in RCCs, in which increased C-11 MET uptake was correlated with high grade (AUC of 0.726 with MET SUV_max_ cut-off > 4.3). By correlating MET SUV_max_ to MTAP expression in RCCs, C-11 MET PET/CT can be used to non-invasively select patients with decreased MET salvage pathways in vivo.

Fuhrman’s grade is a well-known predictor of aggressive RCC behaviour [40,41,42]. However, non-invasive preoperative predictors for the advanced-stage disease are limited. In the present prospective study, we included clinicopathologic and imaging parameters of F-18 FDG and C-11 MET PET/CT to predict advanced-stage disease according to AJCC (pathological TNM stages III and IV). In the univariate analysis, tumour size, F-18 FDG SUV_max_, F-18 FDG MTV, C-11 MET SUV_max_, C-11 MET MTV, and Fuhrman grade were predictive factors. Among these, F-18 FDG MTV was the only independent factor statistically significant in the multivariate analysis. With a cut-off MTV value > 21.3 cm^3^, the advanced-stage disease could be predicted with an odds ratio of 102.67 (95% CI: 9.69–1087.62). Therefore, patients with high glycolytic tumour volumes should be thoroughly evaluated for potential advanced-stage disease.

The detection sensitivity of F-18 FDG PET/CT for primary tumours is different from its sensitivity to extra-renal metastases. For primary tumours, studies have shown a sensitivity of 47–94% for F-18 FDG PET [43,44,45]. In this study, we also found a moderate sensitivity of 75% for the detection of primary tumours. In contrast, F-18 FDG PET/CT is known to be useful in detecting extra-renal metastases with a pooled sensitivity of 91% (95% CI: 84–96%) since primary tumours with increased F-18 FDG uptake are associated with metastases [7]. In our study, F-18 FDG and C-11 MET PET/CT detected extra-renal metastases in all 10 patients (25%). For lymph nodes (*n* = 3) and lung metastases (*n* = 7), both PET/CT modalities played no additional role compared with contrast-enhanced CT. However, all bone metastases (*n* = 3) were detected on F-18 FDG and C-11 MET PET/CT but were negative or equivocal on conventional imaging, including whole-body bone scintigraphy (WBBS). This was likely attributable to the finding that typical osteolytic bone metastases in RCC might not be well-detected on WBBS [46,47]. Further studies with a larger sample of metastatic lesions are needed to validate the value of F-18 FDG or C-11 MET PET/CT in detecting metastases.

While both F-18 FDG and C-11 MET PET/CT were useful in tumour grading, glycolytic tumour volume on F-18 FDG PET/CT was also predictive of advanced stages. Since it is difficult to predict metabolic features in RCCs based on anatomical imaging modalities or clinical parameters, metabolic PET/CT may play an important role in developing new drugs targeting metabolic reprogramming in RCCs. Automated detection of tumours has recently become feasible even at extremely low effective radiation doses (0.11 mSv) using deep neural networks [48]. Localised imaging of the kidneys could be performed using an ultralow dose of F-18 FDG or C-11 MET using deep learning algorithms to simulate full-dose data. If positive uptake by the primary tumour is observed, a standard dose of radiotracer can be injected, and conventional whole-body imaging can be performed to evaluate potential extra-renal metastases. This protocol might maximise the usefulness of metabolic PET/CT by reducing the cost and unnecessary radiation dose to patients whose primary tumour shows no increase in uptake.

The present study has some limitations. First, various subtypes of RCC, such as chromophobe RCC and papillary RCC, were included in addition to clear cell RCC. However, the number of patients was not sufficient to perform subgroup analysis according to the cell type. Second, survival analysis could not be performed, as enough time had not elapsed since the commencement of the study. Survival analysis in the future will help clarify the prognostic role of metabolic PET/CT in RCC. Finally, given the excellent clinical outcomes, patients with clinical stage I tumours were not included in this study.

## 4. Materials and Method

### 4.1. Study Subjects

This prospective study included patients with suspected RCC classified as clinical stage II or higher on preoperative CT. Altogether, 41 patients underwent F-18 FDG and C-11 MET PET/CT staging between September 2016 and September 2018. The mean interval between F-18 FDG and C-11 MET PET/CT scans was 1.6 ± 3.2 days (range: 0–14 days). In addition, all patients underwent conventional imaging, including abdominal CT, chest CT, and WBBS for conventional initial staging work-up. Thirty-nine patients underwent radical nephrectomy shortly after imaging (median, 5 days; range, 1–23 days), and the remaining two patients underwent biopsy alone. For distant metastases, a biopsy was performed when there was a single lesion or when the lesion was noted at an unexpected location for RCC metastasis. No prior treatment was performed before imaging in any of the patients. Among the 41 patients, one patient was diagnosed with a perivascular epithelioid cell tumour and was thus excluded from the analysis. The remaining 40 patients with confirmed RCC were included in this study. Clinical parameters such as sex, age, and histopathological features, including tumour size, histological subtype, and Fuhrman nuclear grade, were assessed. The Fuhrman grade system was based on a four-point scale from 1 (best prognosis) to 4 (worst prognosis) [49]. Pathological tumour staging was performed according to the 2017 TNM criteria of the AJCC [50]. The present prospective study was approved by the Institutional Review Board of the Yonsei University Health System (project number: 1-2016-0032), and informed consent was obtained from all patients.

### 4.2. PET/CT Imaging Protocol

All patients fasted for at least 6 h before being injected with C-11 MET. PET/CT imaging was performed using a PET/CT scanner (Discovery 710, GE Medical Systems, Milwaukee, USA). Twenty minutes after intravenous injection of 370–555 MBq (10–15 mCi) of C-11 MET, a low-dose CT scan (tube voltage, 120 kV; tube current, auto mA) was performed for attenuation correction and precise anatomical localisation. PET emission scans from the cerebellum to the proximal thigh were acquired for 2 min per bed position in a three-dimensional mode for approximately 20 min. The acquired images were reconstructed with an iterative reconstruction algorithm (VUE point FX-SharpIR, iteration: 2, subset: 16, filter cut off: 5 mm) using CT images for attenuation correction.

For the F-18 FDG PET/CT scan, patients were instructed to fast for at least 6 h, and blood glucose levels ≤ 140 mg/dL were ensured before the injection of 5.5 MBq of F-18 FDG per kg body weight. PET/CT images were acquired using the same PET/CT scanner as the corresponding CT transmission scan protocol for attenuation correction. An emission scan was acquired from the cerebellum to the proximal thigh for 2 min per bed position. The same attenuation correction and image reconstruction protocols were used for the F-18 FDG and C-11 MET images.

### 4.3. PET/CT Image Analysis

C-11 MET and F-18 FDG PET/CT images were registered on contrast-enhanced CT or magnetic resonance (MR) images using imaging software (MIM 6.5; MIM Software Inc.). We used automatic registration provided by the software with manual modification. For the measurement of the standardised uptake value (SUV), a spherical volume of interest (VOI) was drawn over the tumour, and the maximum SUV (SUV_max_) was calculated using the following formula: (decay-corrected activity (kBq)/tissue volume (mL))/(injected F-18 FDG or C-11 MET activity (kBq)/body mass (g)). CT or MR images were used as frequently as possible to avoid activity in the renal calyces, pelvis, and upper ureter on F-18 FDG PET/CT images. MTV was defined as the total tumour volume above an SUV cut-off corresponding to the 97.5th percentile of the contralateral kidney (mean SUV of the contralateral kidney + two standard deviations of the contralateral kidney) in the VOI on both F-18 FDG PET/CT and C-11 MET PET/CT.

The degree of F-18 FDG and C-11 MET uptake in the primary tumour was evaluated visually by two experienced nuclear medicine physicians blinded to the patients’ clinical information. Uptake was considered positive if it was similar to or higher than that of the contralateral kidney cortex. Uptake in a lesion similar to or higher than the uptake in the primary tumour was regarded as distant metastasis. Well-known diagnostic pitfalls related to physiological uptake or benign lesions were considered. Interpreters performed visual analysis independently and reached a consensus after reviewing disagreement cases together. 

### 4.4. Statistical Analysis 

For statistical analyses, patients were classified into two groups according to the TNM staging: limited-stage (stages I and II) and advanced-stage (stages III and IV). Fuhrman grade 1 and grade 2 tumours were considered low-grade tumours, and Fuhrman grade 3 and grade 4 tumours were considered high-grade tumours. Simple and stepwise logistic regression analyses were used to predict advanced-stage disease. Variables for logistic regression analyses included sex, age at diagnosis, tumour size, metabolic parameters (SUV_max_, MTV) measured on F-18 FDG PET/CT and C-11 MET PET/CT, and Fuhrman grade. ROC analysis was used to determine the optimal cut-off value of SUV_max_ and MTV for predicting advanced-stage disease using the Euclidean distance. Correlation analyses were performed among the SUV_max_ values of primary tumours on F-18 FDG and C-11 MET PET/CT, tumour size, and Fuhrman grade. The Mann–Whitney U-test was used to compare the SUV_max_ of F-18 FDG and C-11 MET according to Fuhrman’s grade. ROC analysis was performed to calculate the AUC of F-18 FDG SUV_max_ and the AUC of C-11 MET SUV_max_ for predicting high-grade tumours. The differences between the areas under the two dependent ROC curves were compared using the Delong method. The McNemar test was performed to compare F-18 FDG and C-11 MET uptake of the primary tumours on visual analysis. Inter-observer agreement of visual analysis between two interpreters was calculated. Statistical analyses were performed using IBM SPSS Statistics (version 26.0; IBM Corp., Armonk, NY, USA) or MedCalc Statistical Software (version 19.2.6; MedCalc Software Ltd, Ostend, Belgium). For all tests, two-sided *p*-values < 0.05, were considered statistically significant.

## 5. Conclusions

Both F-18 FDG and C-11 MET PET/CT showed variable uptake in RCCs and were useful in predicting tumours with high Fuhrman grades. In particular, high glycolytic volume on F-18 FDG PET/CT was predictive of advanced-stage diseases. The unique metabolic features of PET/CT may play an important role in developing new drugs targeting metabolic reprogramming in RCC.

## Figures and Tables

**Figure 1 cancers-13-02381-f001:**
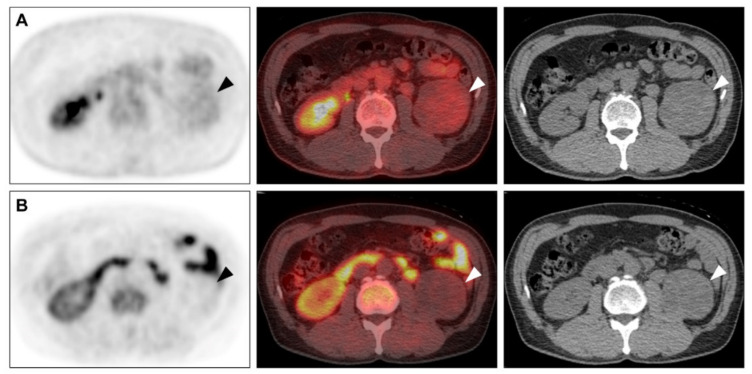
RCC with low MTV. A 56-year-old male had a renal mass (arrowhead) with low F-18 FDG uptake ((**A**) SUV_max_: 2.16, MTV: 0.0 cm^3^) and low C-11 MET uptake ((**B**) SUV_max_: 1.49, MTV: 1.38 cm^3^). A 7.7-cm-sized Fuhrman grade 2 clear cell carcinoma was reported after surgery. The pathological stage was stage II (T2a N0 M0). RCC, renal cell carcinoma; FDG, fluorodeoxyglucose; MET, methionine; SUV_max_, maximum standardised uptake value; MTV, metabolic tumour volume.

**Figure 2 cancers-13-02381-f002:**
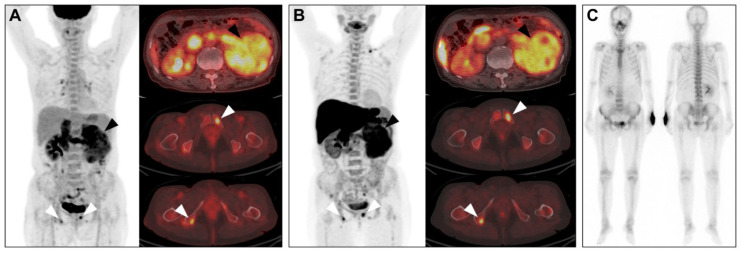
RCC with high MTV. A 66-year-old female had a renal mass (black arrowhead) with high F-18 FDG uptake ((**A**) SUV_max_: 7.64, MTV: 515.38 cm^3^) and high C-11 MET uptake ((**B**) SUV_max_: 9.93, MTV: 424.01 cm^3^). (**C**) Bone metastases (white arrowhead) were also noted on F-18 FDG and C-11 MET PET/CT but were not well-noted on the whole-body bone scan. A 13.0-cm-sized Fuhrman grade 3 clear cell carcinoma was reported after surgery, and the pathological stage was IV (T4 N1 M1). RCC, renal cell carcinoma; FDG, fluorodeoxyglucose; MET, methionine; SUV_max_, maximum standardised uptake value; MTV, metabolic tumour volume.

**Figure 3 cancers-13-02381-f003:**
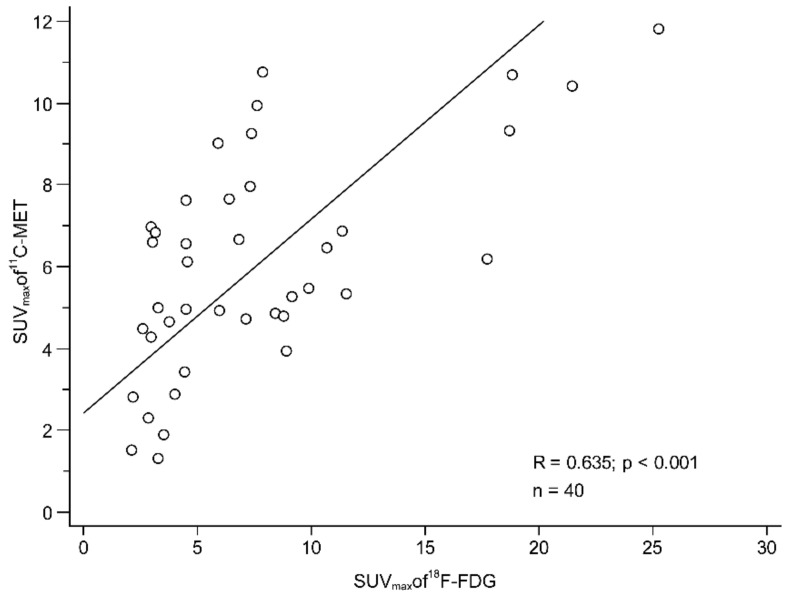
Correlation between F-18 FDG and C-11 MET uptake in RCC. There was a moderate correlation between the SUV_max_ of F-18 FDG and C-11 MET PET/CT (r = 0.63, *p* < 0.001). RCC, renal cell carcinoma; FDG, fluorodeoxyglucose; MET, methionine; SUV_max_, maximum standardised uptake value.

**Figure 4 cancers-13-02381-f004:**
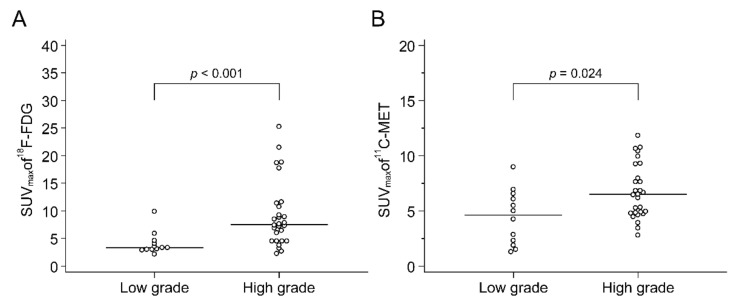
Comparison of F-18 FDG (**A**) and C-11 MET (**B**) uptake of low-grade (grades 1 and 2) and high-grade (grades 3 and 4) RCCs. Tumours with high Fuhrman grades showed higher SUVmax than low-grade tumours on both F-18 FDG ((**A**) 4.1 ± 2.1 vs. 9.2 ± 6.0, *p* = 0.001) and C-11 MET PET/CT ((**B**) 4.4 ± 2.5 vs. 6.7 ± 2.5, *p* = 0.024)). RCC, renal cell carcinoma; FDG, fluorodeoxyglucose; MET, methionine; SUV_max_, maximum standardised uptake value

**Table 1 cancers-13-02381-t001:** Clinical characteristics of the patients.

Characteristics	AJCC Prognostic Groups			
	Stage I (*n* = 5)	Stage II (*n* = 10)	Stage III (*n* = 16)	Stage IV (*n* = 9)
Sex				
Male, *n* (%)	3 (60.0%)	4 (40.0%)	11 (68.8%)	3 (33.3%)
Female, *n* (%)	2 (40.0%)	6 (60.0%)	5 (31.3%)	6 (66.7%)
Age, years, median (Q1–Q3)	61 (58–64)	50 (39–76)	61 (58–67)	62 (56–69)
Tumour size, cm, median (Q1–Q3)	5.4 (4.1–5.8)	8.0 (7.3–8.5)	8.4 (7.1–9.8)	9.0 (8.1–11.5)
Histopathology				
Clear cell type, *n* (%)	3 (60.0%)	8 (80.0%)	14 (87.5%)	9 (100.0%)
Chromophobe type, *n* (%)	1 (20.0%)	2 (20.0%)	1 (6.3%)	0 (0.0%)
Papillary type, *n* (%)	1 (20.0%)	0 (0.0%)	1 (6.3%)	0 (0.0%)
F-18 FDG PET/CT				
SUV_max_, median (Q1–Q3)	3.8 (2.9–4.9)	3.4 (2.9–4.5)	8.0 (5.3–11.5)	7.9 (7.0–12.2)
MTV, median (Q1–Q3)	2.3 (1.1–11.6)	0.6 (0.2–5.5)	109.1 (45.9–194.1)	112.8 (53.9–295.5)
C-11 MET PET/CT				
SUV_max_, median (Q1–Q3)	4.5 (3.9–5.5)	4.9 (2.3–6.1)	6.5 (5.1–7.2)	9.2 (5.1–10.7)
MTV, median (Q1–Q3)	0.0 (0.0–1.4)	3.8 (0.0–9.8)	7.5 (1.4–47.4)	25.8 (9.0–41.4)
Fuhrman nuclear grade				
Grade 1, *n* (%)	0 (0.0%)	0 (0.0%)	0 (0.0%)	0 (0.0%)
Grade 2, *n* (%)	2 (40.0%)	6 (60.0%)	1 (6.3%)	3 (33.3%)
Grade 3, *n* (%)	2 (40.0%)	4 (40.0%)	11 (68.8%)	5 (55.6%)
Grade 4, *n* (%)	1 (20.0%)	0 (0.0%)	4 (25.0%)	1 (11.1%)

AJCC, American Joint Committee on Cancer; FDG, fluorodeoxyglucose; MET, methionine; MTV, metabolic tumour volume; PET/CT, positron emission tomography/computed tomography; Q1, 25th percentile; Q3, 75th percentile; SUV_max_, maximum standardised uptake value.

**Table 2 cancers-13-02381-t002:** Logistic regression results of renal cell carcinoma patients to predict advanced-stage disease (III and IV).

Characteristics	Univariate Analysis		Multivariate Analysis	
	OR (95% CI)	*p*-Value	OR (95% CI)	*p*-Value
Sex (male vs. female)	1.46 (0.40–5.26)	0.568		
Age (years)	1.05 (0.99–1.11)	0.121		
Tumour size (cm)	1.47 (1.03–2.09)	0.033 *		
F-18 FDG PET/CT				
SUV_max_ (≤4.6 vs. >4.6)	26.00 (4.38–154.53)	<0.001 *		
MTV (≤21.3 vs. >21.3)	102.67 (9.69–1087.65)	<0.001 *	102.67 (9.69–1087.65)	<0.001 *
C-11 MET PET/CT				
SUV_max_ (≤5.0 vs. >5.0)	8.71 (2.01–37.76)	0.004 *		
MTV (≤4.2 vs. >4.2)	7.07 (1.68–29.83)	0.008 *		
Fuhrman nuclear grade (1–2 vs. 3–4)	6.00 (1.37–26.20)	0.017 *		

Only one predictor was selected using the stepwise method. CI, confidence interval; FDG, fluorodeoxyglucose; MET, methionine; MTV, metabolic tumour volume; OR, odds ratio; PET/CT, positron emission tomography/computed tomography; SD, standard deviation; SUV_max_, maximum standardised uptake value. * *p*-value < 0.05 was considered significant.

## Data Availability

The data presented in this study are conditionally available with permission from the National Evidence-Based Healthcare Collaborating Agency of South Korea since it was supported by the “Conditional Approval System of Health Technology” funded by the Ministry of Health and Welfare.

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
