# Peer review of "Glycolysis on F-18 FDG PET/CT Is Superior to Amino Acid Metabolism on C-11 Methionine PET/CT in Identifying Advanced Renal Cell Carcinoma at Staging"

_cancers, 2021, doi:10.3390/cancers13102381_

Round 1

Reviewer 1 Report

The revised manuscript has been improved compared to its initial version. However, the limited number of patients examined and the enormous numerical variations in some features measured still remain the Achilles heel that needs further improvement. On the other hand, given its importance in the community and characterized as a pilot study, it can be marginally considered for publication in Cancers high-peer review journal.   

Author Response

Thank you very much for your thoughtful consideration. While we understand your concerns, we hope our study initiates more future research regarding the importance of cancer metabolism and its impact on clinical outcome in patients with RCC.

Reviewer 2 Report

The authors have made good revisions and additions to their submission. I note page 2 (Results) mentions 40 patients while page 9 (Materials and Methods) indicates 41 patients. Could the authors clarify? I recommend a further proof-read for consistency.

Author Response

Thanks for the careful comment. In this study, a total of 41 patients with F-18 FDG and C-11 MET PET/CT was included. One was excluded from the analysis because it was confirmed as a perivascular epithelioid cell tumour. This was previously mentioned on page 9 as below:

  • Page 9, line 280: Among the 41 patients, one patient was diagnosed with a perivascular epithelioid cell tumour and was thus excluded from the analysis. The remaining 40 patients with confirmed RCC were included in this study.

This manuscript is a resubmission of an earlier submission. The following is a list of the peer review reports and author responses from that submission.

Round 1

Reviewer 1 Report

Manuscript in its present form cannot be accepted for publication in CANCERS journal.

Work suffers from the very limited number of patients and the surprisingly high value of heterogeneity. In several cases the standard errors are larger than the respective mean values (e.g. 163.3 +/- 168.5). Obviously, mutational signatures and genetic heterogeneity of cell-clonal populations within RCC tumors are likely responsible for the observed variation in numbers.

Unfortunately, this sounds like a preliminary work that  is not ready yet for clinical exploitation, in the medical management of the disease.   

Reviewer 2 Report

The authors present a prospective study on the usefulness of C11-Methionine and 18F-FDG PET/CT for assessing the aggressiveness of RCCs. The study methodology is sound and the results / discussion appropriate. However, the authors should provide more detail on some parts of the methodology and results sections.

Suggestions for improvement:

  1. The authors should submit, in a tabulated form, the individual SUVmax, MTV, tumour volume / size, tumour characteristics (histopathological type, Fuhrman grade, etc) and patient characteristics for all 40 patients. This is feasible and would provide much needed detail for comparison to previous / future studies.
  2. The authors should clarify how the CTs and MRIs were used to measure tumour size. Has structural volume (based on CT / MRI) been measured or volumetric measurements were made only on PET?
  3. The PET/CT imaging protocol should be more detailed - for example, PET reconstruction parameters should be specified.
  4. 4. What were the registration errors between the PET/CTs and diagnostic CT/MRI? More detail is needed on this process.
  5. The authors state that 2 nuclear medicine physicians reviewed the studies - was this done independently or concomitantly in consensus? can the authors provide inter-observer data and comment on the criteria for the consensus process?